# Undernutrition, Sarcopenia, and Frailty in Fragility Hip Fracture: Advanced Strategies for Improving Clinical Outcomes

**DOI:** 10.3390/nu12123743

**Published:** 2020-12-04

**Authors:** Tatsuro Inoue, Keisuke Maeda, Ayano Nagano, Akio Shimizu, Junko Ueshima, Kenta Murotani, Keisuke Sato, Atsuhiro Tsubaki

**Affiliations:** 1Department of Physical Therapy, Niigata University of Health and Welfare, Shimami-cho 950-3198, Japan; tatsuro-inoue@nuhw.ac.jp (T.I.); tsubaki@nuhw.ac.jp (A.T.); 2Department of Geriatric Medicine, National Center for Geriatrics and Gerontology, Obu 474-8511, Japan; 3Department of Palliative and Supportive Medicine, Graduate School of Medicine, Aichi Medical University, Nagakute 480-1195, Japan; 4Department of Nursing, Nishinomiya Kyoritsu Neurosurgical Hospital, Nishinomiya 663-8211, Japan; aya.k.nagano@gmail.com; 5Department of Nutrition, Hamamatsu City Rehabilitation Hospital, Hamamatsu 433-8127, Japan; a.shimizu.diet@gmail.com; 6Department of Clinical Nutrition and Food Service, NTT Medical Center Tokyo, Tokyo 141-8625, Japan; j.ueshima@gmail.com; 7Biostatistics Center, Kurume University, Kurume 830-0011, Japan; kmurotani@med.kurume-u.ac.jp; 8Okinawa Chuzan Hospital Clinical Research Center, Chuzan Hospital, Matsumoto 904-2151, Japan; keisuke.sato0815@gmail.com

**Keywords:** undernutrition, muscular atrophy, frailty syndrome, fragility hip fracture, elderly

## Abstract

Geriatric patients with hip fractures often experience overlap in problems related to nutrition, including undernutrition, sarcopenia, and frailty. Such problems are powerful predictors of adverse responses, although few healthcare professionals are aware of them and therefore do not implement effective interventions. This review aimed to summarize the impact of undernutrition, sarcopenia, and frailty on clinical outcomes in elderly individuals with hip fractures and identify successful strategies that integrate nutrition and rehabilitation. We searched PubMed (MEDLINE) and Cochrane Central Register of Controlled Trials (CENTRAL) for relevant literature published over the last 10 years and found that advanced interventions targeting the aforementioned conditions helped to significantly improve postoperative outcomes among these patients. Going forward, protocols from advanced interventions for detecting, diagnosing, and treating nutrition problems in geriatric patients with hip fractures should become standard practice in healthcare settings.

## 1. Introduction

Hip fractures are a global public health problem and result in hospitalization, disability, and death [1]. Globally, as the population ages, the number of hip fractures is increasing, and it is expected that 6.3 million people will suffer from hip fracture in 2050 [2]. Hip fracture patients have high mortality [3], experience prolonged disability [4], and require substantial costs for postoperative management [5]. Therefore, management after hip fracture is a critical issue to be resolved.

Hip fracture patients experience multiple geriatric nutritional problems, often including undernutrition, sarcopenia, and frailty at admission, all of which overlap (Figure 1), (Appendix A). These geriatric nutritional problems have significant impacts on disability, the occurrence of complications, and mortality after hip fracture. Therefore, interventions for these factors are a key strategy for improving postoperative clinical outcomes in patients with hip fracture.

Conversely, the effect of interventions for geriatric nutritional problems in patients with hip fracture remains unclear. Nutritional therapy alone was not shown to reduce mortality [6]. Medical professionals often ignore undernutrition, sarcopenia, and frailty, and this unawareness inhibits improvements in clinical outcomes [7]. A focus must be placed on geriatric nutritional problems in hip fracture patients, and effective interventions should be considered. Our review aims to summarize the impact of undernutrition, sarcopenia, and frailty on clinical outcomes and to identify effective interventions combined with nutrition and rehabilitation for hip fracture patients.

## 2. Materials and Methods

### 2.1. Data Sources and Search Strategy

This review adhered to the guidelines of the Preferred Reporting Items for Systematic Reviews and Meta-Analyses (PRISMA) [8]. We searched for relevant literature in PubMed (MEDLINE) and Cochrane Central Register of Controlled Trials (CENTRAL). To review recent studies on undernutrition, sarcopenia, and frailty of patients with hip fracture, we selected observational and intervention studies published in English 10 years since the European Working Group on Sarcopenia in Older People (EWGSOP) was published in 2010 [9]. We used the search terms hip fractures, femoral neck fractures, nutritional status, malnutrition, sarcopenia, muscle atrophy, and frailty.

### 2.2. Study Selection

#### 2.2.1. Inclusion Criteria

The inclusion criteria for the included studies in this review were as follows: (1) Assessment of patients with fragility hip fracture; (2) inclusion of both genders and all races; (3) examination of the impact of undernutrition, sarcopenia, and frailty on clinical outcomes; (4) application of validated nutritional assessments, such as nutritional screening tools, anthropometric parameters, and blood concentrations; (5) evaluation of muscle strength and/or muscle mass for diagnosing sarcopenia; (6) utilization of diagnostic criteria that address multiple factors reflecting vulnerability in the absence of established diagnostic criteria for frailty; (7) clinical outcomes, such as death, complications, hospital stay, discharge disposition, activities of daily living (ADL), mobility, etc.; and (8) observational and intervention study design.

#### 2.2.2. Exclusion Criteria

Editorials, case reports, letters to the editor, review articles, animal studies, and conference abstracts were excluded from this review.

### 2.3. Data Extraction

We extracted the following information from the included studies: Name of the first author, year of publication, country of origin, study design, setting, age, gender prevalence, sample size, screening or assessment tool of nutritional status, diagnostic criteria of sarcopenia and frailty, prevalence of undernutrition, sarcopenia, and frailty, main study outcomes, and main results.

### 2.4. Quality Assessment

We assessed the quality of the included studies using both the National Institutes of Health (NIH) Quality Assessment tool for Observational Cohort and Cross-Sectional Studies and the Quality Assessment of Controlled Intervention Studies [10]. This quality assessment tool comprised 14 items per study design. We scored these items and classified the included studies as “good”, “fair”, or “poor” (Appendix A).

## 3. Undernutrition in Patients with Hip Fracture

### 3.1. Prevalence of Undernutrition

The prevalence of undernutrition with hip fracture is high and varies based on the evaluation tool used, ranging from about 7% to 26% (Table 1). The Mini Nutritional Assessment-Short Form (MNA-SF) [11,12,13,14,15] and the Mini Nutritional Assessment-Full Form (MNA-FF) [12,16,17,18,19] are the most commonly used tools for evaluating nutritional status in patients with hip fracture. The Malnutrition Screening Tool (MST) [20], Controlling Nutritional Status (CONUT) [21,22], Geriatric Nutritional Risk Index (GNRI) [22,23], Malnutrition Universal Screening Tool (MUST) [24], body mass index (BMI) [25,26], serum albumin [12,16,26,27], prealbumin [27], total protein [27], vitamin D [23,27] and lymphocyte count [16] are also used. These evaluation tools are useful for assessing the nutritional status of patients with hip fracture.

### 3.2. Impact of Undernutrition on Clinical Outcomes

A large number of observational studies reported a significant association between undernutrition and clinical outcomes in patients with hip fracture. Most studies set mortality [13,18,19,22,23,24,25,26,28,30] or ADL [11,12,15,17,30] as clinical outcomes and the occurrence of postoperative complications [14,18,21,24], length of hospital stay [18,29], discharge disposition [12,24], readmission [27], mobility [23], and failure after internal fixation [16] as additional outcomes. Inoue et al. [15] and Goisser et al. [17] reported that undernutrition, as evaluated via the MNA-SF and MNA-FF, respectively, was a significant predictor of improved ADL at discharge from acute hospitals and six months postsurgery. Nishioka et al. [11] revealed that improvement in nutritional status via MNA-SF screening during hospitalization in a convalescent hospital was associated with ADL at discharge. Miu and Lam [30] reported that, compared with at-risk and well-nourished patients, malnourished patients screened via the MNA-SF had a higher rate of in-hospital mortality. Gumieiro et al. [28] reported that the MNA-FF score was a predictor of mortality after six months. Vosoughi et al. [25] reported that BMI was an independent risk factor of mortality at one and three years. Conversely, Koren-Hakim et al. [13] reported that the MNA-SF score was not associated with mortality at 36 months. Overall, most of the studies found an association between nutritional status and clinical outcomes in hip fracture patients.

Several studies examining the appropriate nutritional screening tools recommended the use of the MNA-SF for hip fracture patients. The European Society for Clinical Nutrition and Metabolism also recommended the MNA-SF and the Malnutrition Universal Screening Tool and the Nutritional Risk Score 2002 (NRS-2002), which is known as a validated nutritional screening tool [31]. In their comparisons of these validated screening tools, Inoue et al. [32] and Koren-Hakim et al. [33] reported that the MNA-SF was a good predictor of ADL at discharge from an acute hospital, readmission during six months, and mortality at 36 months. In a study comparing the MNA-FF and NRS-2002 [28], only the MNA-FF could predict walking ability and mortality after six months. These results suggested that the use of the MNA-SF or MNA-FF is appropriate for predicting clinical outcomes in patients with hip fracture.

### 3.3. Highlights of Undernutrition in Hip Fracture

Evaluation of nutritional status is important, because undernutrition is a significant risk factor for clinical outcomes in hip fracture patients. The MNA-SF and MNA-FF are the most commonly used tools for nutritional status evaluation and were reported to be significant independent predictors of clinical outcomes. The MNA-SF is a simple and quick nutritional screening tool for nutritional status [34]. Furthermore, calf circumference rather than BMI can be used in the scoring of the MNA-SF, which is an advantage because of the difficulty in accurately measuring body weight on admission for patients with hip fracture. Moreover, the scoring for the MNA-SF includes the following components: functional, psychological, and cognitive aspects. Thus, the MNA-SF can accurately reflect the characteristics of elderly patients with hip fracture and might be the most appropriate nutritional screening tool for clinical outcomes in patients with hip fracture.

## 4. Sarcopenia in Patients with Hip Fracture

### 4.1. Definition of Sarcopenia

Sarcopenia is defined as a muscle disease [35,36] characterized by progressive and generalized decreased muscle strength and loss of muscle mass [9,37]. Sarcopenia is associated with functional disability, death, and other adverse outcomes [7]. Sarcopenia is also associated with osteoporosis [38] and falls [39], therefore, patients with hip fracture are more likely to be sarcopenic.

### 4.2. Prevalence of Sarcopenia

The prevalence of sarcopenia is high in patients with hip fracture. Although the prevalence varies on the basis of the diagnostic criteria, the overall prevalence (for both sexes combined) of sarcopenia ranges from 11% to 76.4% (Table 2). The prevalence ranges from 12% to 81% in males and from 18% to 76% in females. The EWGSOP [9], updated EWGSOP2 [37], Asian Working Group for Sarcopenia (AWGS) [40], and updated AWGS 2019 [41] are often used for diagnosis, and the Foundation for the National Institutes of Health [42,43] and SARC-F [44] were also used in reported studies.

Previous studies reported two ways to diagnose sarcopenia, i.e., using either a combination of muscle strength and muscle mass [45,46,49,50,51,52,53,54,56,57,58,59,60,61,62,63] or muscle mass alone [43,47,48,55]. In all of the studies referenced in the present review, handgrip strength was used to evaluate muscle strength. Dual-energy X-ray absorptiometry [43,46,49,53,54,55,57] and bioimpedance analysis (BIA) [45,51,56,60,62,63] were mostly used to evaluate muscle mass, with computed tomography [47,52] and anthropometric measurement [50,58] also used. Postoperative hip fracture patients have implantation of metal in the lower extremity, and the BIA may overestimate the muscle mass of the operative lower extremity because of its methodological limitations. Therefore, whether BIA is a suitable method for measuring muscle mass in patients with hip fractures is unclear. The criteria for sarcopenia diagnosis are becoming standardized, and further research using standardized diagnostic criteria is necessary in patients with hip fracture.

### 4.3. Impact of Sarcopenia on Clinical Outcomes

Most observational studies reported a significant association between sarcopenia and clinical outcomes in patients with hip fractures. Many studies set outcomes for mortality [48,50,51,52] and ADL [43,46]. Others reported an association between sarcopenia and mobility [50], quality of life (QOL) [53], length of hospital stay [47], discharge disposition [50], and the development of dysphagia [56]. Di Monaco et al. [46] reported the association between sarcopenia and ADL at admission to a convalescent hospital. Landi et al. [43] reported the association between sarcopenia and ADL at discharge from a rehabilitation hospital and after 3 months, and Steihaug et al. [50] reported the association between sarcopenia and mobility after 1 year. Nagano et al. [56] reported an association between sarcopenia and the development of dysphagia after hip fracture. Regarding mortality, Kim et al. [48] reported that sarcopenia was not associated with mortality at one year postoperatively but was associated with mortality at five years. Conversely, Byun et al. [52] reported an association between sarcopenia in women and one-year mortality. Malafarina et al. [51] reported sarcopenia was a predictor of mortality at seven years. Overall, sarcopenia was found to be a significant independent predictor of postoperative clinical outcomes, and the diagnosis of sarcopenia is important to improve clinical outcomes.

### 4.4. Highlights of Sarcopenia in Hip Fracture

The prevalence of sarcopenia is very high, and sarcopenia is a significant predictor of adverse outcomes in patients with hip fractures. The diagnostic criteria of the EWGSOP, updated EWGSOP2, AWGS, and updated AWGS 2019 are mainly used for the diagnosis of sarcopenia, depending on the race of the patients. The use of standardized diagnostic criteria has had a positive impact on the increase in sarcopenia research in patients with hip fracture. However, sarcopenia is often overlooked in clinical practice [7], and there are no intervention studies in hip fracture patients with sarcopenia. Thus, this type of study in hip fracture patients with sarcopenia is strongly needed.

## 5. Frailty in Patients with Hip Fracture

### 5.1. Definition of Frailty

Frailty is defined as a state of vulnerability accompanied by various preliminary reductions in the ability to maintain or regain homeostasis when exposed to stressors [64]. However, no standardized diagnostic criteria of frailty exist, and various tools were used in reported studies [64,65,66]. A previous study reported an association between frailty and the incidence of hip fractures [67], with a large proportion of hip fracture patients expected to have frailty.

### 5.2. Prevalence of Frailty

The diagnosis of frailty in patients with hip fracture is hindered by a lack of standardized diagnostic criteria for frailty. These criteria vary in the studies referenced in the present review (Table 3). Therefore, discussing the prevalence of frailty in hip fracture patients is difficult. The most commonly used criteria are the frailty phenotype reported by Fried et al. [68] and the frailty index reported by Rockwood et al. [69]. Frailty phenotype has the following five features or criteria: Weakness, slow gait speed, low physical activity, exhaustion, and unintentional weight loss [68]. Frailty is diagnosed if a positive score is obtained for three or more symptoms or signs out of the five criteria.

The concept of the frailty index consists of the accumulation of health-related deficits, such as signs, symptoms, disease, and disability. The frailty index is easy to use in clinical practice because it consists mainly of medical conditions [69] and can be evaluated from the medical record. Patel et al. [70], Inoue et al. [78], Vasu et al. [76], and Pizzonia M et al. [82] adopted 19 items and Krishnan et al. [71] adopted 51 items to develop the modified frailty index for hip fracture patients. These models suggest that frailty is a continuous score that considers disability, comorbidity, and symptoms. Higher scores are considered to be associated with greater frailty. They reported an association of the modified frailty index with mortality [70,76,82], occurrences of complications [78], length of hospital stay [71], discharge disposition [71,78] from acute hospital, and low functional recovery [78]. Further studies are needed to enable an easy diagnosis of frailty in clinical practice for hip fracture patients.

### 5.3. Impact of Frailty on Clinical Outcomes

Many previous studies reported that frailty was a predictor of adverse outcomes. The clinical outcomes included mortality [70,71,74,75,76,77,82,84], the occurrence of complications [72,73,78], length of hospital stay [72,73,74], ADL [78,80,81,83], QOL [79], and discharge disposition [72,74,78,83]. However, few well-designed studies were conducted. Thus, it is necessary to develop diagnostic criteria that are simple, highly accurate, able to predict adverse outcomes, and suitable for hip fracture patients.

## 6. Nutritional Intervention for Patients with Hip Fracture

Based on the current evidence, the effectiveness of nutritional therapy alone for hip fracture patients is unclear. A systematic review [6] of nutritional interventions for hip fracture patients reported only low-quality evidence to reduce complications and no clear effect on mortality. Many intervention studies examined the effect of oral administration of protein [85,86,87,88,89,90,91,92], β-hydroxy-β-methylbutyrate [93], vitamin D [94,95,96], whey protein [97,98], or combined calcium β-hydroxy-β-methylbutyrate (CaHMB), vitamin D, and protein intake [99] on clinical outcomes. One randomized controlled trial for hip fracture patients conducted an intervention to calculate energy requirements by measuring the resting energy expenditure using an indirect calorimeter [100]. In individual randomized controlled trials, the group that received the nutritional intervention had better outcomes than the control group in terms of occurrence of complications [87,100], severity of pressure ulcers [88], length of hospital stay [89], readmission rate [94], nutritional status [86], muscle strength [98], muscle mass [91,93], and wound-healing period [99]. Conversely, there was no significant difference in nutritional status [85,89] or mortality [87] between the group that received a nutritional intervention alone and the control group. The effects of nutritional intervention on ADL are not consistent [87,89,90,91,98]. There were no intervention studies that reported enhanced rehabilitation used in combination with nutritional therapy. These discrepancies might suggest that nutritional interventions alone are insufficient to improve clinical outcomes.

## 7. Combined Nutritional Intervention with Rehabilitation Exercise

A combination of nutrition and exercise interventions is effective for elderly patients with sarcopenia. A combination of amino acid intake and exercise improved muscle strength, muscle mass, and ADL of community-dwelling women with sarcopenia [101] and sarcopenic patients with cerebrovascular disease [102]. A meta-analysis reported that the combination of nutrition and exercise had a positive effect on physical function in community-dwelling elderly individuals [103]. Combined nutrition and exercise interventions promoted muscle protein synthesis compared with each of these interventions alone [104]. Thus, these combination interventions for hip fracture patients may contribute to improved clinical outcomes.

## 8. Advanced Strategies for Improvement of Clinical Outcomes

To improve clinical outcomes effectively, medical professionals should be aware of geriatric nutritional problems in hip fracture patients (Figure 2). On the basis of geriatric nutritional evaluation, we must be careful about iatrogenic sarcopenia [7]. Iatrogenic sarcopenia is caused by hospitalization and is drug-related [7]. Hospitalization-related iatrogenic sarcopenia is caused by physicians, nurses, and other medical professionals [105,106]. Iatrogenic sarcopenia mainly comprises inactivity- and nutritional-related factors. Inactivity-related iatrogenic sarcopenia is mainly caused by unnecessary inactivity during the perioperative period. In hospitalized hip fracture patients, approximately 99% of the day consists of sedentary time [107]. The incidence of sarcopenia in acute hospitals is approximately 15%, and the duration of bed rest is associated with the incidence of sarcopenia [108]. In patients in rehabilitation hospitals, increased time away from bed is more effective in improving ADL [109]. Medical professionals should pay close attention to iatrogenic sarcopenia, and avoiding unnecessary bed rest, immobility, and deconditioning in patients could prevent activity-related sarcopenia.

In hip fracture patients, nutritional-related iatrogenic sarcopenia requires a comprehensive approach. Only 17.5% of patients meet their energy requirements in the first week after hip surgery [110]. Additionally, multiple factors are associated with reduced food intake after fractures [111,112], and it is clear that interventions that merely administer supplements are insufficient for improving clinical outcomes. Bell et al. [113] reported that intensive individualized, multidisciplinary (orthopedic and geriatric physician, nursing staff, physiotherapists and occupational therapists, dietitian, pharmacist, etc.) interventions reduced barriers to food intake; food intake increased in the group with multidisciplinary intervention (mean 1489.0 kcal/day, protein intake of 1.13 g/body weight) compared with the group with conventional care (mean 707.4 kcal/day, protein intake of 0.60 g/body weight) in hip fracture patients. Additionally, medical professionals should pay attention to sarcopenic dysphagia accompanied by deterioration in nutritional status after hip surgery [56]. A multidisciplinary, comprehensive pragmatic intervention trial is required for hip fractures with overlapping undernutrition, sarcopenia, and frailty. Compared with randomized controlled trials, pragmatic trials can be routinely conducted with less stringent inclusion and exclusion criteria. Therefore, selection bias can be controlled, and the results can be easily generalized to routine clinical practice. Comprehensive multidisciplinary interventions are necessary to prevent nutritional-related iatrogenic sarcopenia in patients with hip fracture.

## 9. Comprehensive Intervention Based on Combined Nutritional Intervention with Rehabilitation Exercise for Patients with Hip Fractures

The geriatric nutritional evaluation, a comprehensive approach that combines nutritional management and rehabilitation, is a key strategy for improving clinical outcomes [105,106,114]. The concept of “rehabilitation nutrition” [114] invented in Japan may be effective for managing geriatric nutritional problems in fragility hip fracture patients. “Rehabilitation nutrition” is defined as that which (i) holistically evaluates the presence and causes of nutritional disorders, sarcopenia, and excess or deficiency of nutrient intake as per the International Classification of Functioning, Disability and Health; (ii) conducts rehabilitation nutrition diagnosis and rehabilitation nutrition goal setting; and (iii) elicits the highest body functions, activities, participations, and QOL by improving nutritional status, sarcopenia, and frailty using “nutrition care management in consideration of rehabilitation” and “rehabilitation in consideration of nutrition” in people with a disability and frail older people [114]. This rehabilitation nutrition concept can maximize functional recovery and QOL through the diagnosis and intervention of undernutrition, sarcopenia, and frailty. Previous studies reported the usefulness of this comprehensive approach, which combines nutritional management and rehabilitation [102,115]. Future research on comprehensive interventions combined with nutrition and rehabilitation, specifically for hip fracture patients, is strongly needed.

## 10. Strengths and Limitations

The strength of this review is that we summarized recent research that focused on the nutritional problem of elderly patients with hip fracture and mentioned new intervention strategies for geriatric nutritional problems. However, this review also has methodological limitations. For example, we did not use a strict literature search for a systematic review, which is necessary to further explore the impact of sarcopenia and frailty on the clinical outcomes of hip fractures.

## 11. Conclusions

The overlap between undernutrition, sarcopenia, and frailty is a characteristic of fragility hip fracture patients. Geriatric nutritional problems have a strong impact on adverse outcomes after hip fracture. To improve clinical outcomes effectively, medical professionals should be aware of geriatric nutritional problems in hip fracture patients. A comprehensive approach that combines nutritional management and rehabilitation is a key strategy for improving clinical outcomes. New, comprehensive, advanced, and hip-fracture-specific intervention strategies are strongly needed.

## Figures and Tables

**Figure 1 nutrients-12-03743-f001:**
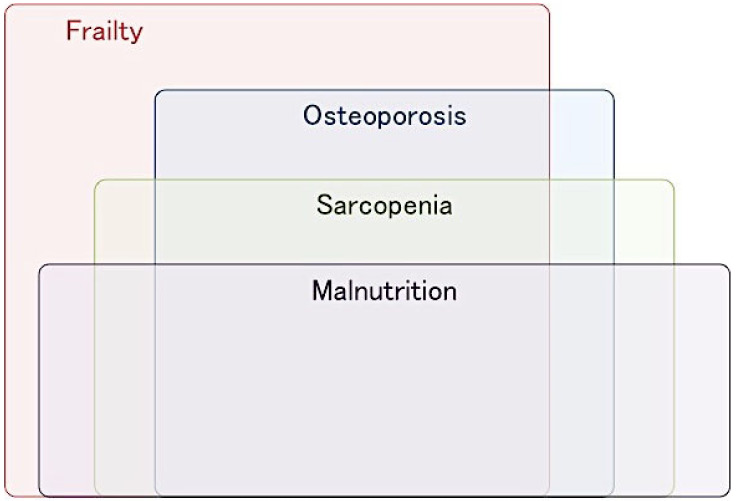
The overlapping geriatric nutritional problems in patients with fragility hip fracture.

**Figure 2 nutrients-12-03743-f002:**
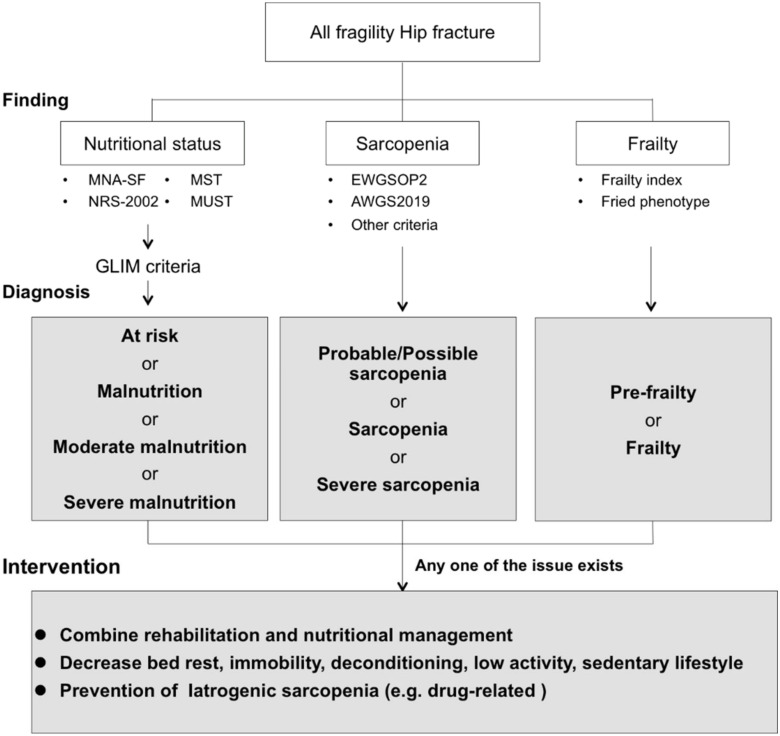
The specific strategies of geriatric nutritional evaluation and advanced intervention for patients with fragility hip fracture. Abbreviations: MNA-SF, Mini Nutritional Assessment-Short Form; MST, Malnutrition Screening Tool; NRS-2002, Nutrition Risk Screening 2002; MUST, Malnutrition Universal Screening Tool; EWGSOP, European Working Group on Sarcopenia in Older People; AWGS, Asian Working Group for Sarcopenia.

**Table 1 nutrients-12-03743-t001:** Assessment of nutritional status, prevalence of undernutrition, and the impact of undernutrition on clinical outcomes in patients with hip fracture.

Author, Year, Country	Design, Setting	Age (Years) Male/Female, *n* (%)	Sample Size	Evaluation Tool (Timing of Assessment)	Prevalence of Undernutrition	Outcome	Main Results
Miyanishi et al., 2010 [26] Japan	Observational study, acute hospital	Mean 79 24 (18.9)/103 (81.1)	129	Serum albumin BMI	Not stated	Four-year mortality	In, multiple logistic regression analysis, serum albumin level (OR 5.854, *p* < 0.001) and BMI (OR 1.169, *p* = 0.02) significantly influenced mortality.
Koren-Hakim et al., 2012 [13] Israel	Observational study, acute hospital	Mean 83.5 (SD 6.0) 61 (28.4)/154 (71.6)	215	MNA-SF (at admission and up to 48 h after admission)	Well-nourished: 44.2% At risk: 44.2% Malnourished: 11.6%	In-hospital complications Mortality (up to 36 months)	Only comorbidity and low functioning can predict long-term mortality (a minimum of 12 up to 36 months). Nutritional status had no effect on outcomes.
Gumieiro et al., 2012 [28] Brazil	Prospective observational study, general hospital	Mean 80.2 (SD 7.3) 20 (23.3)/66 (76.7)	86	MNA-FF NRS-2002 (within the first 72 h of the patient’s admission)	Not stated	Gait status (patients who could walk or could not walk) and mortality at 6 months after hip fracture	In a multivariate analysis, only the MNA-FF was associated with gait status (OR 0.773, 95% CI 0.663−0.901) and mortality 6 months after hip fracture (HR 0.869, 95% CI 0.757−0.998).
Drevet et al., 2014 [29] France	Prospective observational study, university hospital	Mean 86.1 (SD 4.4) 15 (30)/35 (70)	50	MNA-FF (no details provided)	At risk for PEM: 58% PEM: 28%	Activities of daily living Hospital stay	PEM was associated with functional dependence (*p* = 0.002) and 8 days longer mean hospital stay (*p* = 0.012).
Goisser et al., 2015 [17] Germany	Prospective observational study, urban maximum care hospital	Mean 84 (SD 5) (21)/(79)	97	MNA-FF (preoperative nutritional status was evaluated retrospectively)	At risk: 38% Malnourished: 17%	Barthel Index after 6 months	Malnourished patients suffered more from remaining losses in ADL ≥25% of initial Barthel Index points (*p* = 0.033), and regained their prefracture mobility level to a lesser extent (*p* = 0.020) than well-nourished patients.
Bajada et al., 2015 [16] UK	Retrospective observational study, general hospital	Mean 79 years (range: 60–96 years) 19 (18)/89 (82)	108	Serum albumin (normal level > 35 g/L) Lymphocyte count (normal 1−4.5 × 109 L) (on admission)	No details provided	Failure of internal fixation	In binary logistic regression analysis, lymphocyte count, and albumin levels were independent predictors of failure of internal fixation.
van Wissen et al., 2016 [18] Netherlands	Retrospective cohort study, acute hospital	Mean Malnourished: 85 (SD 5) At risk: 84 (SD 5) Well-nourished: 83 (SD 5) 61 (27.0)/165(73.0)	226	MNA-FF (before surgery)	Well-nourished: 4.9% At risk: 26.5% Malnourished: 68.6%	Hospital stay Postoperative complications, Mortality (in-hospital and 1-year)	Preoperative malnutrition is associated with in-hospital (OR 4.4; 95% CI 1.0, 20.4) and 1-year mortality (OR 2.7; 95% CI 1.1, 7.0). Malnutrition was not associated with any other outcome.
Miu et al., 2017 [30] China	Observational study, rehabilitation unit	Mean 83.5 (SD 7.5) 74 (33.9)/44 (66.1)	218	MNA-SF (within 72 h of admission)	Well-nourished: 21.1% At risk: 52.6% Malnourished: 26.1%	Functional status and place of residence at 6 months Hospital stay Mortality (in-hospital, 6 months)	Functional recovery was slower in the malnourished group. In-patient mortality was higher in malnourished patients than in those at risk of malnourishment and well-nourished individuals.
Helminen et al., 2017 [12] Finland	Prospective observational study, acute hospital	No details provided 169 (28.5)/425 (71.5)	594	MNA-SF MNA-FF Serum albumin (preoperative period)	MNA-SF Well-nourished: 53% At risk: 40% Malnourished: 7% MNA-FF Well-nourished: 35% At risk: 58% Malnourished: 7% Serum albumin <34 g/L: 46%	Poorer mobility (transfer to more assisted living accommodation) Mortality (1 month, 4 months, and 1 year after fracture)	Risk of malnutrition and malnutrition measured by MNA-FF predicted mobility and living arrangements within 4 months of hip fracture. At 1 year, risk of malnutrition predicted mobility and malnutrition predicted living arrangements when measured by the MNA-FF. Malnutrition, but not risk measured by the MNA-SF, predicted living arrangements at all time points. Neither measure predicted 1-month mobility.
Vosoughi et al., 2017 [25] Iran	Cross-sectional study, university hospital	Mean 75.7 (SD 10.6) 318 (43.9)/406 (56.1)	724	BMI (at admission)	No details provided	Mortality at 3 months and 1 year	Multivariate logistic regression analysis recognized age (OR 1.08; 95% CI 1.05, 1.11), BMI (OR 0.88; 95% CI 0.82−0.96), and smoking (OR 1.76; 95% CI 1.05−2.96) as major independent risk factors for 1- and 3-year mortality.
Mazzola et al., 2017 [14] Italy	Prospective observational study, university hospital	Mean 84.0 (SD 6.6) 106 (25.5)/309 (74.5)	415	MNA-SF (within 24 h of admission)	Well-nourished: 36.6% At risk: 44.6% Malnourished: 18.8%	Postoperative delirium	Multivariate regression analysis showed that those at risk of malnutrition (OR 2.42; 95% CI = 1.29–4.53) and those overtly malnourished (OR 2.98; 95% CI = 1.43–6.19) were more likely to develop postoperative delirium.
Inoue et al., 2017 [15] Japan	Prospective observational study, three acute hospitals	Mean 82.7 (SD 9.2) 69 (10.1)/165 (80.9)	204	MNA-SF (first few days after admission before surgery)	Well-nourished: 27.0% At risk: 48.0% Malnourished: 25.0%	FIM at discharge	In multiple regression analyses, MNA-SF was a significant independent predictor for FIM at discharge (well-nourished vs. malnourished, β = 0.86, *p* < 0.01).
Nishioka et al., 2018 [11] Japan	Retrospective observational cohort study, convalescent rehabilitation units	Mean 85 years (21.8)/(78.2)	110	MNA-SF (on admission and at discharge)	Only malnourished patients at admission were included	FIM at discharge Discharged to home	Multivariable analysis revealed a significant association between improvement in nutritional status and higher FIM score at discharge (β = 7.377, *p* = 0.036). No association with discharge to home.
Stone et al., 2018 [27] USA	Retrospective observational cohort study, acute hospital	Not stated 241(39.7)/366(60.3)	607	Albumin Prealbumin Total protein Vitamin D	Not stated	Thirty-day readmission	The model incorporated four nutritional makers (albumin, prealbumin, total protein, and vitamin D) with an internally cross-validated C-statistic of 0.811 (95% CI; 0.754, 0.867).
Zanetti et al., 2018 [19] Italy	Observational study, acute hospital	Mean 84.7 (SD 7.4) 259 (21.4)/952 (78.6)	1211	MNA-FF (within 72 h from admission)	Mean MNA-FF score: 22.3 (SD 5.1)	Three, six and twelve-month mortality	Poor nutritional status was significantly associated with 3, 6, and 12- month mortality after adjustment for potential confounders.
Kotera et al. 2019 [22] Japan	Retrospective observational cohort study, acute hospitals	Mean 87 (SD 6) Not stated	607	GNRI CONUT	GNRI Normal: 0.8% Light: 3.0% Moderate: 5.7% Severe: 14.4% CONUT Normal: 1.6% Light: 2.7% Moderate: 8.1% Severe: 38.9%	Mortality of 180 days	The GNRI value in the nonsurvivors was significantly lower than that in the survivors. The CONUT value in the nonsurvivors was significantly higher than that in the survivors.
Yagi et al., 2020 [21] Japan	Retrospective observational cohort study, community-based hospital	Median 86 years (interquartile range 80–90) (19.9)/(80.1)	211	CONUT (admission day)	Malnourished (CONUT score >1): 78.7%	Postoperative complications	Multivariable analysis found that the CONUT score was an independent risk factor for postoperative complications (OR 1.21; 95% CI = 1.01–1.45).
Hao et al., 2020 [23] USA	Retrospective observational study (secondary analysis), 47 sites in North America	Mean 82 (SD 7) (27)/(73)	290	25-hydroxyvitamin D GNRI (preoperative)	25-hydroxyvitamin D (ng/mL) ≥30: 17% 20 to <30: 37% 12 to <20: 34% <12: 12% GNRI No risk: 33 Some risk: 33 Major/moderate risk: 34	Mortality and mobility at 30 and 60 days after surgery	Compared with patients with <12 ng/mL, those with higher 25(OH)D concentrations had higher rates of walking at 30 days (*p* = 0.031): 12 to <20 ng/mL (adjusted OR 2.61; 95% CI 1.13, 5.99); 20 to <30 ng/mL (3.48; 1.53, 7.95); ≥30 ng/mL (2.84; 1.12, 7.20). There was also greater mobility at 60 days (*p* = 0.028) in patients with higher 25(OH)D compared with the reference group (<12 ng/mL). GNRI <92 showed an overall trend to reduce mobility (adjusted *p* = 0.056) at 30 but not at 60 days. There was no association of vitamin D or GNRI with mortality at either time.
Han et al., 2020 [24] UK	Retrospective observational study, National Health Service hospital	Mean 83.8 (SD 8.6) 349(28.2)/890(71.8)	1239	MUST	Low risk Medium risk High risk	Mobilization (starting rehabilitation within 1 day after surgery) Pressure ulcers In-patient mortality Change in discharge destination	Compared with the well-nourished group, malnourished individuals showed increased risk for failure to mobilize within 1 day of surgery (OR 1.6; 95% CI 1.0–2.7), pressure ulcers (OR 5.5, 95% CI, 1.8–17.1), in-patient mortality (OR 2.3; 95% CI, 1.1–4.8), and discharge to residential/nursing care (OR 2.8; 95% CI, 1.2–6.6).

Abbreviations: BMI, body mass index; OR, odds ratio; SD, standard deviation; MNA-SF, Mini Nutritional Assessment-Short Form; MNA-FF, Mini Nutritional Assessment-Full Form; NRS-2002, Nutrition Risk Screening 2002; CI, confidence interval; HR, hazard ratio; FIM, Functional Independence Measure; PEM, protein energy malnutrition; ADL, activities of daily living; GNRI, Geriatric Nutritional Risk Index; CONUT, Controlling Nutritional Status; MUST, Malnutrition Universal Screening Tool.

**Table 2 nutrients-12-03743-t002:** Diagnosis criteria of sarcopenia, prevalence, and its impact on clinical outcomes in patients with hip fracture.

Author, Year, Country	Design, Setting	Age Male/Female, *n* (%)	Sample Size	Diagnosis Criteria Measurement Methods of Muscle Strength, Muscle Mass, Physical Function	Prevalence of Sarcopenia	Outcome	Main Results
González-Montalvo et al., 2015 [45] Spain	Prospective observational study, university hospital	Mean 85.3 (SD 6.8) 47 (20.3)/382 (79.7)	479	EWGSOP Handgrip strength Bioimpedance analysis	17.1%	Barthel Index at discharge	In the multivariate analysis, sarcopenia was not associated with functional prognosis at discharge (OR 1.68, 95% CI 0.99–2.84).
Di Monaco et al., 2015 [46] Italy	Observational study, rehabilitation hospital	Normal: 78.9 (SD 7.7) Presarcopenia: 73.8 (SD 5.5) Sarcopenia: 81.3 (SD 7.5) All female: 138 (100)	138	EWGSOP Handgrip strength Dual-energy X-ray absorptiometry	Presarcopenia: 17% Sarcopenia: 58%	Barthel Index (at admission, end of the rehabilitation course)	Sarcopenia was associated with Barthel Index scores at the time of assessment but not at the end of the rehabilitation course after adjusting for multiple adjustments (*p* < 0.001).
Landi et al., 2017 [43] Italy	Observational study, Geriatric Rehabilitation Unit of the hospital	Mean age 81.3 (SD 4.8) 45 (36.4)/82 (64.6)	127	FNIH Dual-energy X-ray absorptiometry	Sarcopenia: 48%	Barthel Index (at discharge and 3 months after discharge)	After adjustment for potential confounders, participants with sarcopenia had a significantly increased risk of incomplete functional recovery compared with nonsarcopenic patients (OR 3.07, 95% CI 1.07–8.75).
Di Chang et al., 2018 [47] Taiwan	Retrospective observational study, university hospital	Mean age 81.1 (SD 12.2) 24 (26.4)/67 (73.6)	91	Computed tomography (total skeletal muscle area at L4)	No details provided	Hospital stay Perioperative mortality Medical complications In-hospital blood transfusion volume Readmission rate at 90 days	Low skeletal muscle index was independently associated with longer length of hospitalization (*p* = 0.032) but was not associated with any other outcomes.
Kim et al., 2018 [48] Korea	Retrospective observational study, National Police Hospital	Mean 78.5 years (range, 65–94 years) 27 (29.7)/64 (70.3)	91	Choi et al. reported criteria Computed tomography (L3)	49.5%	One-year and five-year mortality rates	Kaplan–Meier analysis showed that sarcopenia did not affect the 1-year mortality rate (*p* = 0.793) but had a significant effect on the 5-year mortality rate (*p* = 0.028). Both perioperative sarcopenia (*p* = 0.018) and osteoporosis (*p* < 0.001) affected the 5-year mortality rate.
Yoo et al., 2018 [49] Korea	Retrospective observational study, university hospital	Mean 77.8 (SD 9.7) 78 (24.1)/246 (75.9)	324	AWGS Handgrip strength Dual-energy X-ray absorptiometry	37.7%	One-year mortality	Osteosarcopenia (15.1%) was higher for 1-year mortality than other groups (normal: 7.8%, osteoporosis alone: 5.1%, sarcopenia alone: 10.3%).
Steihaug et al., 2018 [50] Norway	Prospective observational study, acute hospital (three hospitals)	Mean 79.4 (SD 8.2) (24)/(76)	282	EWGSOP Handgrip strength The formula reported by Heymsfield et al. (using gender, height, arm circumference, and triceps skinfold) New Mobility Score	38%	Change in New Mobility Score Resident of a nursing home Death	Sarcopenia did not predict change in mobility (*p* = 0.6), but it was associated with having lower mobility at 1-year (*p* = 0.003), becoming a resident of a nursing home (OR 3.2, *p* = 0.048), and the combined endpoint of becoming a resident of a skilled nursing home or death (OR 3.6, *p* = 0.02).
Malafarina et al., 2019 [51] Spain	Prospective observational study, two rehabilitation units	Mean 85.2 (SD 6.3) 49 (26.2)/138 (73.8)	187	EWGSOP2 Handgrip strength Bioimpedance analysis 4 meter walking test	Incident sarcopenia during hospitalization: 54 patients Sarcopenia at admission and at discharge (chronic sarcopenia): 41 patients Sarcopenic at admission but reverted sarcopenia during the admission period (reverted sarcopenia): 17 patients	Mortality after 7 years	Cox regression analyses showed that sarcopenia was a risk factor for mortality (HR: 1.67, 95% CI 1.11–2.51) and low handgrip strength (HR: 1.76, 95% CI 1.08–2.88).
Byun et al., 2019 [52] Korea	Retrospective study, university hospital	Mean 78.4 (SD 9.7) 121 (24.5)/373 (75.5)	494	AWGS Handgrip strength Computed tomography (psoas cross-sectional area at L4–L5 level)	No details provided	One-year mortality	After adjusting for potential confounders, the lowest quintile of psoas cross-sectional area was significantly associated with mortality only in females (HR 1.76, 95% CI 1.05–2.70).
Chen et al., 2020 [53] Hong Kong	Prospective observational study, acute hospital	Mean 80.72 (SD 9.66) 36 (25.9)/103 (74.1)	139	AWGS Handgrip strength Dual-energy X-ray absorptiometry	50.36%	EQ5D and Barthel Index at 6 months after the operation	After 6 months, patients with sarcopenia had a poor Barthel Index and a lower EQ5D than patients without sarcopenia before injury.
Chiles Shaffer et al., 2020 [54] USA	Prospective observational study, the seventh cohort of the Baltimore Hip Studies	Male: 81.0 (SD 7.5) Female: 80.2 (SD 7.6) 82 (51.3)/78 (48.7)	160	EWGSOP IWGS FNIH Handgrip strength Dual-energy X-ray absorptiometry Gait speed	No details provided	Sarcopenia prevalence over 12 months after hip fracture	Sarcopenia prevalence was stable over time in men by all definitions, whereas the prevalence in women by FNIH was lowest at 2 months, significantly increased at 6 months (*p* = 0.03) and remained higher at 12 months. Sarcopenia prevalence differed significantly by sex and varied by time point and definition; however, when different, men had a higher prevalence than women did (*p* < 0.05).
Shin et al., 2020 [55] Korea	Retrospective cohort study, university Hospital	Mean age 74.1 (range, 25–96) 35 (25.9)/100 (74.1)	135	AWGS Dual-energy X-ray absorptiometry	45.9%	Harris Hip Score Parker’s mobility score at the last follow-up Discharge disposition	In multiple regression analysis, no significant association was found between sarcopenia and the Harris Hip Score of mobility at the last follow-up, nonunion, or time to union.
Nagano et al., 2020 [56] Japan	Retrospective observational study, acute hospital	Mean 85.9 (SD 6.5) All female patients, 89 (100)	89	AWGS 2019 Handgrip strength Bioimpedance analysis	76.4%	Incidence of dysphagia on day 7 and discharge	All patients who developed dysphagia had underlying sarcopenia.
Ha et al., 2020 [57] Korea	Cross-sectional study, acute hospital	Not sarcopenia: 76.02 (SD 6.87) Sarcopenia: 82.62 (SD 7.72) 22 (19.1)/93 (80.9)	115	SARC-F, EWGSSOP2, AWGS, IWGS Handgrip strength Dual-energy X-ray absorptiometry	SARC-F: 63.5% EWGS2: 43 (37.4%) AWGS: 35 (30.4%) IWGS: 60 (52.2%)	Comparison of the results with criteria	Accuracy of SARC-F was that the sensitivity, specificity, positive predictive value, negative predictive value, and positive predictive value with the EWGSOP2 criteria as the reference standard were 95.35%, 56.94%, 56.94%, 95.35%, and 71.3%, respectively.

Abbreviations: EWGSOP, European Working Group on Sarcopenia in Older People; OR, odds ratio; CI, confidence interval; FNIH, Foundation for the National Institutes of Health; AWGS, Asian Working Group for Sarcopenia; IWGS, International Working Group on Sarcopenia; HR, hazard ratio.

**Table 3 nutrients-12-03743-t003:** Diagnosis criteria of frailty and its prevalence and impact on clinical outcomes in patients with hip fracture.

Author, Year, Country	Design, Setting	Age Male/Female, *n* (%)	Sample Size	Diagnosis Criteria Details of Criteria	Prevalence of Frailty	Outcome	Main Results
Patel et al., 2014 [70] USA	Retrospective observational study, acute hospital	Mean 81.05 (SD 8.45) No gender details provided	697	Modified frailty index 19 items Comorbidities, cognitive function, and walking ability	No details provided	One-year and two-year mortality rates after femoral neck fracture	Patients with a modified frailty index had an OR of 4.97 for 1-year mortality and an OR of 4.01 for 2-year mortality as compared with patients with an index less than 4.
Krishnan et al., 2014 [71] UK	Prospective study, university-affiliated community hospital	Mean 81 (range, 47–101) 47 (26.5)/131 (735)	178	Frailty index Fifty-one deficits Motivation, self-rated health, cognitive assessments, clock face drawing, comorbidities, continence, mobility, and functional independence Low-frailty group (FI ≤ 0.25), intermediate (FI > 0.25–0.4), high-FI group (FI > 0.4)	Low-frailty group (FI ≤ 0.25): 56 (31.5%) Intermediate (FI >0.25–0.4): 58 (32.5%) High (FI >0.4): 64 (36%)	Hospital stay Discharge disposition	The mean length of hospital stay for the intermediate group was 36.3 days in the high-FI group compared with 67.8 days in the high-FI group (*p* < 0.01). 30-day mortality was 3.4% for the intermediate group compared with 17.2% for the high-FI group (*p* < 0.001).
Kistler et al., 2015 [72] USA	Prospective observational study, university-affiliated community hospital	Mean 86 (SD 4) 6 (17)/29 (83)	35	Fried frailty index (modified for a post fracture population) Shrinking, exhaustion, slowness, weakness, and physical activity Participants with a total score of 3 or higher were considered frail	51%	Overall hospital complication rate Hospital stay Complications	Frail patients (67%) versus nonfrail patients (29%) had a complication (*p* = 0.028). Mean length of stay was longer in patients with frailty (7.3 (SD) 5.9 vs. 4.1 (SD) 1.2 days, *p* = 0.038).
Gleason et al., 2017 [73] USA	Retrospective observational study, acute hospital	Mean 82.3 (SD 7.4) 44 (25.1)/131 (74.9)	175	The FRAIL scale Five-question assessment Fatigue, resistance, aerobic capacity, illnesses, and loss of weight Classified the patients into three categories: robust (score = 0), prefrail (score = 1–2), and frail (score = 3–5)	Robust (*n* = 29): 16.6% Prefrail (*n* = 73): 41.7% Frail (*n* = 73): 41.7%	Postoperative complications Unplanned intensive care unit admission Hospital stay Discharge disposition 30-day readmission and mortality	There was a statistically significant association between frailty and both length of stay (4.2, 5.0, and 7.1 days, *p* = 002, in robust, prefrail, and frail groups) and the development of any complication (3.4%, 26%, and 39.7%, *p* = 0.03) after surgery. There were also significant differences in discharge disposition (31% of robust vs. 4.1% frailty, *p* = 0.008) and follow-up completion (97% of robust vs. 69% of frail).
Choi et al., 2017 [74] Korea	Retrospective study, university hospital	Mean 80.4 (IQR 75.3–85.3) 139 (28.8)/343 (71.3)	481	Hip-Multidimensional Frailty Score Sex, Charlson Comorbidity Index, Albumin, Koval grade, risk of falling, MNA, and mid-arm circumference High risk: >8 and low risk: ≤8	High risk: 24.3%	One-year all-cause mortality Postoperative complication Hospital stay Institutionalization	High-risk patients showed a higher risk of six-month mortality (HR: 3.545, 95% CI: 1.466–8.572) than low-risk patients after adjustment. Hip-Multidimensional Frailty Score could predict six-month mortality, postoperative complications, and prolonged hospital stay after surgery. Hip-Multidimensional Frailty Score more precisely predicted six-month mortality than age or existing tools (*p* values of comparison of ROC curve: 0.002, 0.004, and 0.044 for the ASA classification, age, and NHFS, respectively).
Winters et al., 2108 [75] Netherlands	Retrospective observational cohort study, general hospital	Mean 83.0 (SD 6.6) 71 (25)/215 (75)	280	Groningen Frailty Indicator questionnaire Consisted of 15 questions Physical, cognitive, social, and psychological impairments Score on a scale of 0–15 Score of 4 or higher suggests frailty VeiligheidsManagementSysteem Three items (cognitive impairment or confusion during earlier admissions, falls in the last 6 months, and physical impairments) Falling and another question to determine the frailty	Groningen Frailty Indicator questionnaire: 60% VeiligheidsManagementSysteem: 58%	Mortality 3-years and 30 days after surgery	VMS showed a statistically significant difference in overall survival as compared to nonfrail patients (57 vs 80%, respectively, *p* < 0.001) with an HR of 3.5 (95% CI 2.1–5.7; *p* < 0.001)). Classification according to GFI yielded a lower but still significant HR 2.3 (95% CI 1.2–4.1; *p* = 0.008).
Vasu et al., 2018 [76] India	Retrospective observational study, acute hospital	Not stated 34 (56.7)/26 (43.3)	60	Modified frailty index Nineteen items Comorbidities, cognitive function, and walking ability	Mean modified frailty index score: 3	90 days mortality	Modified frailty index and 90-day mortality showed a significantly direct correlation, with <0.001.
Chen et al., 2019 [77] Taiwan	Prospective observational cohort study	≤75: 34.3% 76–85: 41.2% ≥86: 25.5% 79 (32.2)/166 (67.8)	245	Chinese-Canadian Study of Health and Aging Clinical Frailty Scale Ranged from 1 (very fit) to 7 (severely frail).	Robust: 31.4%. Prefrail: 46.1% Frail: 22.4%	1, 3, and 6-month postoperative emergency department visits Readmissions Mortality	More cumulative events occurred for frail than for robust patients for each adverse outcome. Frailty had a long-term effect on each adverse outcome.
Inoue et al., 2019 [78] Japan	Retrospective observational study, two acute hospitals	Mean 83.7 (SD 7.4) 52 (19.3)/217 (80.7)	274	Modified frailty index Nineteen items Comorbidities, cognitive function, and walking ability	Mean modified frailty score: 3.2 ±1.9 points (minimum to a maximum range of 0 to 9)	Efficiency on the motor-Functional Independence Measure Postoperative complication Discharge disposition	Higher modified frailty index was significantly associated with increased likelihood of lower functional recovery (OR, 1.60; 95% CI, 1.32–1.93), occurrence of postoperative complication (OR, 1.32; 95% CI, 1.13–1.54) and not returning home (OR, 1.77; 95% CI, 1.38–2.26).
Van De Ree et al., 2019 [79] Netherlands	Prospective observational study, 10 participating Dutch hospitals	Mean 80.27 (SD 8.62) 206 (29.6)/490 (70.4)	696	Groningen Frailty Indicator questionnaire Consisted of 15 questions Physical, cognitive, social, and psychological impairments Score on a scale of 0–15 Score of 4 or higher suggests frailty	53.3%	EuroQol-5 Dimensions ICEpop CAPability measure for Older people	Frailty was negatively associated with EuroQol-5 Dimensions (β −0.333; 95% CI −0.366 to −0.299), self-rated health (β −21.9; 95% CI −24.2 to −19.6), and capability and well-being (β −0.296; 95% CI −0.322 to −0.270) 1 year after hip fracture.
Jorissen et al., 2020 [80] Australia	Retrospective cohort study, historical national cohort of the Registry of Senior Australians	Mean 85.8 (SD 6.3) 1164 (24.4)/3607 (75.6)	4771	Frailty index Forty-four deficits Eight activity limitations, 24 health conditions, and three signs and symptoms 0–0.18 (quartile 1), 0.19–0.23 (quartile 2), 0.24–0.27 (quartile 3), and 0.28–0.41 (quartile 4)	Quartile 1: 1307 (27.4%) Quartile 2: 1158 (24.3%) Quartile 3: 1123 (23.5%) Quartile 4: 1183 (24.8%)	2 year survival ADL limitations Permanent residential aged care for patients living in the community	The two-year survival of patients following hip fracture was 43.7% (95% CI 40.9–46.7%) in those in the highest quartile of frailty, compared with 54.4% (95% CI 51.8–57.2%) for those in the lowest quartile (HR = 1.25, 95% CI 1.11–1.41). No associations were found between pre-fracture frailty and post fracture ADL limitations. No association of frailty with transition to permanent residential aged care for patients living in the community was observed (HR = 0.98, 95% CI 0.81–1.18).
Lu et al., 2020 [81] China	Longitudinal and observational study, university hospital	Mean 77.5 (SD 8.5) 43 (33)/87 (67)	130	The modified Krishnan FI Physical health, mental health, cognitive function, self-care ability, life satisfaction, and social function The Canadian study of health and aging frailty index Cognition, existing diseases, self-care deficits, and abnormal physical signs	The modified Krishnan FI Low: 39% Medium: 50% High: 12% The Canadian study of health and aging frailty index Low: 63% Medium: 36% High: 0.8%	Death Rate of readmission to the hospital Fall within 3 months Hip function Daily activities at 3 months after surgery	The modified Krishnan FI correlated with the Japanese Orthopedic Association hip score (pain, activity, walking ability, and ability for daily living; *R* = 0.249, *p* = 0.005), whereas the Canadian study of health and aging frailty index was not correlated (*R* = 0.125, *p* = 0.170). Both the modified Krishnan FI (*R* = 0.415, *p* < 0.001) and the Canadian study of health and aging frailty index (*R* = 0.332, *p* < 0.001) were significantly correlated with the functional recovery scale score.
Pizzonia et al., 2020 [82] Italy	Prospective observational study, acute hospital	Mean 86.5 (SD 5.65) 80 (22)/284 (78)	364	Modified frailty index 19 items Comorbidities, cognitive function, and walking ability	Robust: 2.2% Prefrail: 14.9% Frail: 82.9%	Mortality (median follow-up of 2.4 years)	Modified frailty index was predictive of long-term mortality.
Low et al., 2020 [83] Australia	Prospective cohort study, rehabilitation and two geriatric evaluation and management wards	Median 86 years (interquartile range 81–90) 254 (30.1)/590 (69.9)	844	Clinical Frailty Scale 9 points scale	69.9%	FIM efficiency Mobility Discharge disposition	Clinical Frailty Scale was the strongest independent predictor of poorer FIM efficiency, inability to recover pre-fracture mobility, and return to community dwelling.
Narula et al., 2020 [84] Australia	Retrospective observational study, acute hospital	Nonfrail: 73.8 (8.8) Vulnerable: 80.3 (9.0) Mildly frail: 84.3 (8.3) Moderately frail: 84.7 (6.9) Severely frail: 86.6 (7.3) 135 (26.5)/374 (73.5)	509	Clinical Frailty Scale 9 points scale	Non frail: 15.7% Vulnerable: 17.9% Mildly frail: 23.0% Moderately frail: 13.8% Severely frail: 29.7%	30 day and 1-year mortality	The Clinical Frailty Scale demonstrated superior discriminative ability in predicting mortality (area under the curve 0.699; 95% CI 0.651 to 0.747) when compared with the ASA and chronological age groups.

Abbreviations: CI, confidence interval; FI, frailty index; IQR, interquartile range; ASA, American Society of Anesthesiologists; NHFS, Nottingham Hip Fracture Score; MNA, Mini Nutritional Assessment; ROC, receiver-operating characteristic; FIM, Functional Independence Measure; ADL, activities of daily living; OR, odds ratio; HR, hazard ratio.

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
