# Peer review of "Undernutrition, Sarcopenia, and Frailty in Fragility Hip Fracture: Advanced Strategies for Improving Clinical Outcomes"

_nutrients, 2020, doi:10.3390/nu12123743_

Round 1

Reviewer 1 Report

This is a well written and very comprehensive review on an important subject. 

The concepts are well described and the conclusions are clear.

I have only two remarks, that I suggest requires a short discussion section:

  • this is a narrative review, could the authors describe how they selected the literature? If this is not possible, I would encourage a discussion on both the strengths and limitations of this review- the absence of a systematic search being one of them
  • the other issue is the quality of the used studies. Could the authors explain how the general quality of the studies was? and how should potential bias that was found regularly or frequently, be avoided in future studies? and what studies are needed now? 

Reviewer 2 Report

This manuscript presents an overview of the highly clinically relevant, and often entirely disregarded overlap between undernutrition, sarcopenia, and frailty in hip fracture.

However, the lack of adherence to the PRISMA checklist, and absence of a methodology component within the manuscript, combined with omission of a number of key manuscripts.

I would be happy to review a future version that applies a systematic approach to critiquing and presenting a summary of the evidence to date. This should be considerate of, and preferably aligning to, the relevant checklist (eg. PRISMA for Nutrition Journal).

Round 2

Reviewer 2 Report

Thank you for including the methodology section and reference to PRISMA.

However, there are still a number of outstanding omissions that preclude me undertaking a thorough review.

refer PRISMA checklist under sections:

Title; abstract; registration;search; data collection processes; study selection.

The lack of provision of full electronic search strategy and number of studies screened, assessed for eligibility, etc with a flow diagram precludes me from being able to duplicate your methodology. This is important as there are still a number of key references that are not included, for example in Table 1.
